# Leveraging River Network Topology and Regionalization to Expand SWOT-Derived River Discharge Time Series in the Mississippi River Basin

Cassandra Nickles [1,*] and Edward Beighley [1,2]

1 Department of Civil and Environmental Engineering, Northeastern University, Boston, MA 02115, USA; r.beighley@northeastern.edu
2 Department of Marine and Environmental Sciences, Northeastern University, Boston, MA 02115, USA
* Correspondence: nickles.c@northeastern.edu

**Abstract:** The upcoming Surface Water and Ocean Topography (SWOT) mission will measure rivers wider than 50–100 m using a 21-day orbit, providing river reach derived discharges that can inform applications like flood forecasting and large-scale hydrologic modelling. However, these discharges will not be uniform in time or coincident with those of neighboring reaches. It is often assumed discharge upstream and downstream of a river location are highly correlated in natural conditions and can be transferred using a scaling factor like the drainage area ratio between locations. Here, the applicability of the drainage area ratio method to integrate, in space and time, SWOT-derived discharges throughout the observable river network of the Mississippi River basin is assessed. In some cases, area ratios ranging from 0.01 to 100 can be used, but cumulative urban area and/or the number of dams/reservoirs between locations decrease the method's applicability. Though the mean number of SWOT observations for a given reach increases by 83% and the number of peak events captured increases by 100%, expanded SWOT sampled time series distributions often underperform compared to the original SWOT sampled time series for significance tests and quantile results. Alternate expansion methods may be more viable for future work.

**Keywords:** discharge; drainage area ratio; regionalization; urbanization; hydrology; remote sensing; surface water and ocean topography

## 1. Introduction

The upcoming Surface Water and Ocean Topography (SWOT) satellite, expected to launch in 2022, will enable discharge estimation in large rivers (>50–100 m wide) globally [1]. The satellite will utilize a 21-day repeat orbit-cycle within ±78 degrees latitude, observing the latitude near the poles more often than near the equator per cycle [2]. SWOT observations will be unevenly distributed in space/time throughout the 21-day cycle. For example, if a location is measured three times in the orbit-cycle, this does not indicate it will be measured every seven days (i.e., the observations could be on days 8, 19, and 20). In addition, locations near each other may not be observed an equal number of times or on the same days, resulting in some portions of a river being measured more or less frequently than others [3]. Only 55% of the Dartmouth Flood Observatory's recorded flood events would have been directly captured by SWOT because of its orbit-cycle [4]. With such irregularity, it would be ideal to efficiently translate discharge from a SWOT observed portion of a river to another unobserved portion to expand the number of measurements at a given location. Note, satellite remote sensing is not a replacement for in-situ gauge networks or global modeled discharge estimates, but rather a compliment to them, having the potential to expand overall data collection [5]. To further optimize the usefulness of SWOT satellite data in particular, methods for spatial and temporal measurement expansion need to be assessed. One approach for spatiotemporal measurement expansion

is regionalization. Many cite their reasoning for regionalization as the need to estimate discharge in ungauged [6] or even politically ungauged regions [7], but it can also be used for time series expansion broadly, especially in the case of remotely sensed measurements.

The transfer of data or information from a known portion to an unknown portion of a river or catchment is broadly labeled as regionalization. Often, the density of hydrologic information is not uniform throughout a river basin. Thus, expanding regional knowledge through spatial relationships improves the ability to capture hydrologic dynamics. Regionalization is often used to estimate river discharge or flow statistics in an ungauged region. Understanding discharge dynamics in a region enables water resources managers to more accurately allocate water [8], manage flood risks [9], determine design capacities for civil infrastructure [10], inform hydrologic models [11], and even assess climate change impacts [12]. As regional hydrologic information shifts in part due to land cover change [13], urbanization [14], and man-made hydrologic infrastructure [15], it is important to understand how they impact common methods for propagating hydrologic information throughout a river network.

Reviews for estimating discharge in ungauged catchments often organize regionalization into two methods: a direct regionalization of flow and flow metrics, and the regionalization of model parameters [16]. Many regionalization studies involve hydrologic models, and readers are referred to [17] for a comprehensive review. Different types of hydrologic information are often regionalized, including flow duration curves [18,19], model parameters [20,21], peak flow [22], low flows [23], annual flow statistics [24], and daily discharge [25]. A variety of methods to transfer spatial information have been used, including but not limited to: the drainage area ratio method [22], regression models [18], nonlinear scaling [26], spatial proximity [21], physical similarity [27], map-correlation methods [25], multi-model approaches [28], data assimilation [29], and recurrent neural networks [30]. Some techniques are relatively simple, only needing a scaling factor to operate, while others are much more complex in their approach. For the context of this paper, the drainage area ratio method, denoted as one of the oldest and most common regionalization methods [22,25,31,32], is assessed.

The drainage area ratio method is an efficient, fundamental approach, with its simplest form being:

$$Q_u = Q_g \left( \frac{A_u}{A_g} \right) \tag{1}$$

where $Q$ indicates discharge, $A$ is drainage area, and the subscripts g and u are gauged and ungauged sites, respectively. The method began before remote sensing river products were in use, so in this case, "gauged" can also mean "remotely sensed." It is known for its ease of use; drainage area is not difficult to obtain for all catchments globally. This method is ideal because instead of estimating discharge using a complex method requiring many datasets, it uses nature's own model for discharge in an unknown region: a neighboring catchment's record of discharge [33].

The drainage area ratio method is indeed convenient, but often there is a large variability in its accuracy. Fourteen sites in a 68 km$^2$ forested boreal catchment are tested [34] finding no matter the time scale: daily, weekly, monthly, seasonally, and even annually, results using the method vary greatly. In the context of two small basins in Virginia [31], the drainage area ratio method works relatively well if the sites have similar hydrologic and low-flow characteristics, yet poorly in comparison to regression analysis. Four estimation techniques are observed for low flows in 182 gauged sites in the south-eastern United States [23], one category being scaling methods. They admit the drainage area ratio as a scalar performs so poorly, it is not even included in their results. Although the drainage area ratio method has a mixed record of accuracy, it is still used widely today.

The drainage area ratio method is in widespread use by analysts and managers of surface-water resources [35,36], though no clear spatial limit on the drainage area ratio currently exists. Many studies have reported conflicting ratios where the method has been successful. One suggests that transposition of discharge (transferring estimates from a

gauged location to a nearby location on the same river) is of value within the area ratio of 0.75 to 1.25 after assessing its potential for peak flow estimation across 439 gauges with drainage areas less than 1000 km$^2$ in the southeast United States [22]. Another recommends selecting a reference stream gauge with a drainage area ratio between 0.5 and 1.5 in rural regions of Idaho with drainage areas between 5 and 32,000 km$^2$ [37], while yet another reports 0.3 to 1.5 as an acceptable range from gauges in Massachusetts whose drainage areas range from 13 to 550 km$^2$ [38]. A ±50% drainage area limit has been utilized as well [32], citing numerous publications analyzing peak flows and even low flows while ultimately admitting conclusive evidence for this ±50% rule has not been obtained. In addition, the success of the drainage area ratio has been reported in a small, forested basin in New York (2900 km$^2$), using drainage area ratios as low as 0.005 [39], though admitting the lack of urban areas may contribute greatly to its success. Regionalization methods either work well or fail miserably, and often it is difficult to find the limiting factors and underlying mechanics of why [20].

One factor contributing to its failure in some cases is that the drainage area ratio method operates under the major assumption that discharge scales linearly with drainage area. This implies all parts of the basin contribute nearly the same volume of water at nearly the same rate as either runoff or as recharge to the water table. Realistically, this is not often the case, though scaling relationships between discharge and drainage area have been found to have strong correlations [24,26]. All parts of the basin may not contribute the same volume of water at the same rate as either runoff or as recharge to the water table. Urbanization increases overland flow and provides anthropogenic pathways, decreasing infiltration and increasing the rate water is delivered into main channels [14,40]. Man-made instream infrastructure also directly challenges the assumption. Reservoirs, dams, locks, hydropower plants, etc. all obstruct the natural flow of rivers [41], something a drainage area ratio scaling factor would not consider.

Currently, selecting a reliable donor gauge for transfer of information to another location remains a challenge for hydrologists because no standard guidance regarding selection criteria exists [36]. Many studies have analyzed the drainage area ratio method on smaller regional scales [34,39] where little anthropogenic disruptions impacting flow occur. In the case of remotely sensed measurements like those that will be gleaned from the SWOT satellite, avoiding regions with anthropogenic influence is nearly impossible. Historically, large cities and man-made river obstructions have often been built along large rivers, and SWOT observed portions of rivers will most likely have catchments greater than roughly 1000–2000 km$^2$ (i.e., an area large enough to produce rivers at least 50–100 m wide). Could a scaling factor method like the drainage area ratio method be effectively used on SWOT observable rivers to expand hydrologic knowledge? What would be the limitations on its use? Is there a limit to the drainage area ratio range that is effective? How much influence do cities and dams have on the accuracy of such a method? Building on these questions, the objectives of this paper are: (1) to assess the assumption that discharge scales linearly with drainage area in large river basins, (2) to identify limitations (e.g., ratio limits, presence of reservoirs, or degree of urbanization) for the drainage area ratio method's optimal use, and (3) to apply these limits to expand the spatial and temporal extent of SWOT derived discharges in the Mississippi River basin.

## 2. Materials and Methods

### 2.1. The Surface Water and Ocean Topography (SWOT) Mission Irregular Orbit Cycle

Operating a Ka-band SAR interferometric (KaRIn) system with two 50 km swaths and a nadir track in between, SWOT's spatial resolution far surpasses all previous altimetry satellites, widening the applicability for discharge products' use globally. Because of the SWOT mission's irregular 21-day orbit cycle, however, a given river segment could be observed at a differing temporal frequency than its neighboring river reaches upstream or downstream. Figure 1 shows an example of the spatial and temporal irregularities for a portion of SWOT observable rivers in and around Illinois in the United States. This figure is

not meant to define the study region but rather tangibly display why SWOT discharge time series expansion is needed. An example swath representing a SWOT pass is highlighted in blue underneath the river network. In this region, SWOT will observe the river reaches a minimum of zero and maximum of four times during the 21-day orbit cycle.

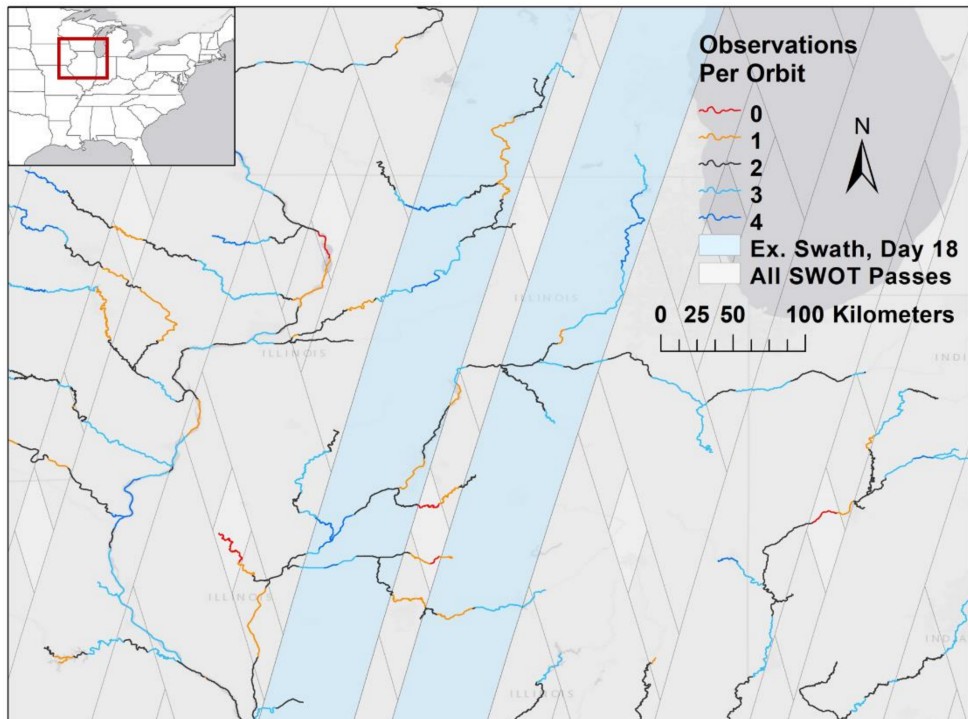

**Figure 1.** An example of SWOT observable rivers near the lake Michigan (dark gray) and Illinois area of the United States. The rivers are colored by the number of observations per 21-day orbit cycle per 10 km reach. SWOT swaths are visible in the background with a specific swath in blue to better visualize the layout of a single pass on day 18 of the cycle. Each pass consists of two 50 km swaths.

It is conceivable that a resulting time series from any portion of these rivers could be easily expanded with satellite measurements from a neighboring reach that is observed more frequently or at different times. Therefore, observations could be transferred from one site to the other using the drainage area ratio method constrained by limitations for its optimal use. Here, the following questions are explored: On a river segment, how does the number of SWOT observed days in a 21-day cycle change as measurements are added through analyzing river network connectivity? Does the increase in observations improve results if analyzed with the same methods used previously [3] to compared discharge frequency distributions and peak flows? Specific methods to answer these questions are described in Section 2.4.

### 2.2. Assessing the Drainage Area Ratio Method

The drainage area ratio method, as defined in the Introduction (Equation (1)), needs only a source location's discharge and drainage area and the drainage area of the locations where discharge is to be estimated. The discharge of an unknown location is found by multiplying the donor gauge's discharge by the ratio of the drainage areas. In this study, there are not any truly "unknown" locations, but results of the drainage area ratio method discharge are compared against observed discharges for a subset of in-situ gauges. The error metric utilized to determine how results compare is the Kling-Gupta Efficiency (KGE) [42,43]. KGE is calculated between two gauge time series: the truth time series from

the location desired, and the donor gauge time series multiplied by the drainage area ratio scaling factor. KGE from [43] is defined as:

$$\text{KGE} = 1 - \sqrt{(R-1)^2 + (\beta - 1)^2 + (\gamma - 1)^2} \qquad (2)$$

$$R = \frac{cov(Qd, Qo)}{\sigma_{Qd}\sigma_{Qo}} , \quad \beta = \frac{\mu_{Qd}}{\mu_{Qo}} , \quad \gamma = \frac{\sigma_{Qd}/\mu_{Qd}}{\sigma_{Qo}/\mu_{Qo}} \qquad (3)$$

where $R$ is the correlation coefficient, $\beta$ is the bias ratio, and $\gamma$ is the relative variability between two time series. In this case, $Q_d$ is the donor discharge and $Q_o$ is the observed discharge. $\mu$ indicates the mean and $\sigma$ indicates the standard deviation of the specified time series. A KGE of one indicates that the drainage area ratio scalar multiplied by the donor gauge time series gave exactly the time series of the desired location. A KGE of $-0.41$ is the threshold for the metric where the model improves upon the mean flow benchmark, unlike the commonly used Nash-Sutcliffe Efficiency (NSE), whose threshold is zero [44]. KGE combines a breadth of factors hydrologists often analyze: Pearson's correlation coefficient, bias, and the spread of data.

Citation [44] recommends analyzing individual KGE components to better understand the holistic value as there is not one general threshold for a "good" KGE that will fit all studies. Dissecting the metric by its three factors, a good KGE value for the purposes of the study is defined. Pearson's correlation coefficient is considered "high" if the value lies between 0.5 and 1. For the bias and relative variability terms, expected error metrics for the SWOT satellite are referenced. Citation [45] shows the median relative bias for their discharge algorithm's performance is $-17\%$ and their median relative root-mean-square error (rRMSE) is 42% across 19 studied rivers. In this study, simulated SWOT time series are derived using these metrics (as described in Section 2.4). For consistency and knowing that relative variability, like rRMSE, is indicative of the spread of data, a "good" bias ratio is defined as 0.83, and a "good" relative variability threshold is defined as 0.58, the ratios that match these expected SWOT errors. Using a correlation coefficient of 0.5, a bias ratio of 0.83, and a relative variability of 0.58, a threshold for a "good" KGE value of 0.32 is obtained.

Once KGE is calculated for all pairs, results are analyzed and characteristics in each location that make the drainage area ratio successful or unsuccessful in its estimation are determined. A reasonable KGE would be greater than $-0.41$, improving upon the mean flow benchmark, and as discussed above, a good KGE is defined here as above 0.32. Specifically, the anthropogenic factors observed that may be influencing results are whether the river is obstructed by man-made infrastructure between the gauges and the difference in urban drainage area between the two gauges. The major sources of urban area are often larger scale cities along rivers. According to the U.S. census, the top 200 cities in population have a median size of 196 km$^2$ [46]. Thus, as an indication of potential urbanization influence on discharge, an allowable urban land area threshold between two gauges is selected as 196 km$^2$. The anthropogenic influences having been characterized, the criteria for optimal use of the drainage area ratio method are identified and the spatial limits of its success on the Mississippi River Basin are quantified.

### 2.3. Data and Study Region

The Mississippi River and catchment network used in this study is a vectorized subset of MERIT Hydro [47,48], limited to river reaches with drainage areas greater than 1000 km$^2$, a likely SWOT observable reach [1]. The river reaches are approximately 10 km in length as defined by MERIT Hydro vectors. On these SWOT observable rivers, discharge measurements from 373 USGS gauges are gleaned. These gauges are a subset of the 453 gauges used previously [3] but parsed down to 373 well-established gauges that have a record of at least 20 years. The analysis is performed on three years of daily gauge discharge from April 16, 2013 to April 15, 2016. The 3-year period is selected to coincide with previous analysis [3] and the expected SWOT mission lifetime. The period 2013–2016 begins with El Niño Southern Oscillation (ENSO)-neutral hydrologic years (2013–2015) and

then strong El Niño years (2015–2016), increasing rainfall in the region. Only one major flood occurred in the basin coming through the Ohio Valley in January 2016. Drainage area ratios between each gauge are calculated to determine which could be potential donor gauges, utilizing a wide range of possible ratios from 0.01 and 100. With this ratio range, 354 of the 373 gauges are eligible to be a donor gauge (Figure 2a). The drainage area ratio values tested have an equal amount of upstream (<1) and downstream (>1) donations tested (Figure 2b). For example, if a downstream gauge has 100% more drainage area than an upstream gauge, the drainage area ratio to donate downstream from one to the other would be 2, while to donate upstream, the ratio would be 0.5. Approximately half of the downstream drainage area donations possible have drainage area ratios between 1 and 10, and the remaining half have ratios between 10 and 100. For upstream donations, half of the gauge comparisons have drainage area ratios between 0.01 and 0.1, while the other half span 0.1 to 1. The range of drainage areas for the gauges covers four orders of magnitude (Figure 2c), with the minimum drainage area being 1100 km$^2$, the median 12,800 km$^2$, and the maximum being approximately 1.8 M km$^2$.

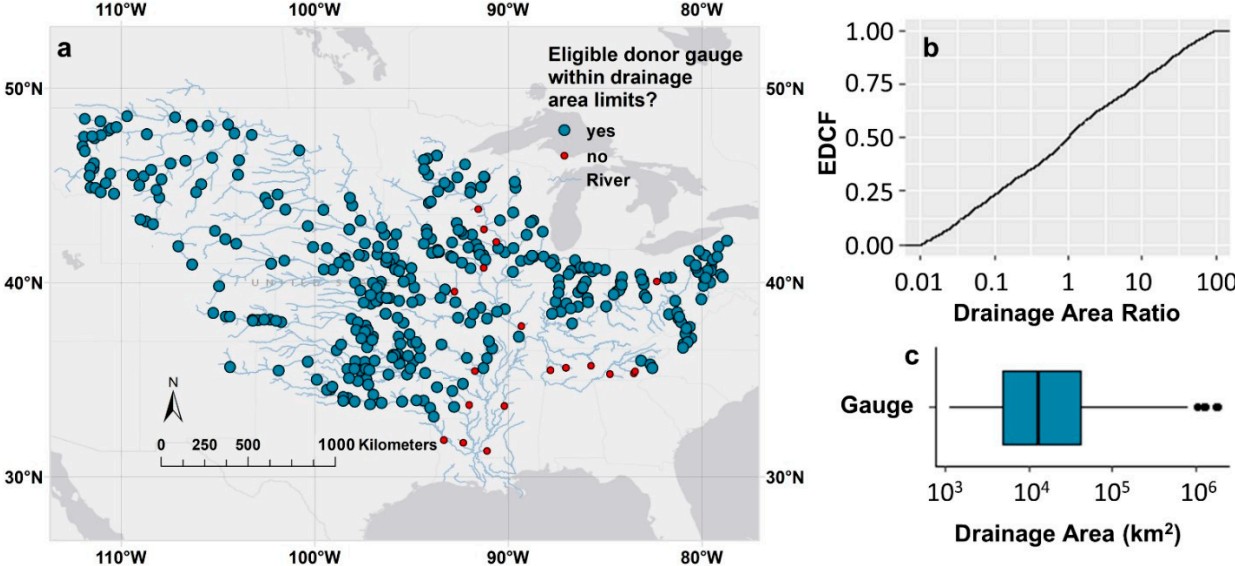

**Figure 2.** (**a**) 373 USGS gauges in the Mississippi River Network with rivers reaches that have drainage areas greater than 1100 km$^2$. The gauges are symbolized by their eligibility to be a donor gauge, meaning they have a drainage area ratio greater than 0.01 or less than 100 with another gauge. (**b**) Histogram and empirical cumulative distribution function (ECDF) of the participating donor gauges' drainage area ratios with those they donate to. There are 4344 interactions possible within the range from 354 gauges. (**c**) Boxplot of donor gauge drainage areas using the log scale.

To determine the urbanization per catchment, MODIS Land Cover Type CMG Yearly L3 Global 0.05 Degree data for 2019 (i.e., MCD12C1 v006) [49] is utilized. The "Urban and Built-up Lands" layer gives a percentage of urban area per 0.05 degree of land area. To locate river obstructions such as dams and reservoirs, the Global River Obstruction Database (GROD)'s "Dams" class is used to map 811 Dams in the United States [41]. GROD gives latitude and longitude points for its infrastructure, therefore, to better visualize the major reservoirs in GIS, polygon shapefiles are also used from the Global Reservoir and Dam Database (GRanD) [50], which contains 206 of the 811 GROD dams. Figure 3 shows the MODIS urban land cover layer utilized and the locations of dams/reservoirs throughout the basin. The Mississippi River basin drains nearly three million km$^2$ of drainage area and extends from the Rocky Mountains and semi-arid prairies in the north-west to the humid regions near the Great Lakes and the subtropical lowlands near the Gulf of Mexico. The regions contain a mix of highly agricultural areas and sprawling industrial cities. The Mississippi River's elevation starts at 450 m above sea level in northern Minnesota and

drops to sea level at the Gulf of Mexico. Depending on the location, the mean annual precipitation in the Mississippi River Basin ranges from 200–1600 mm/year [51].

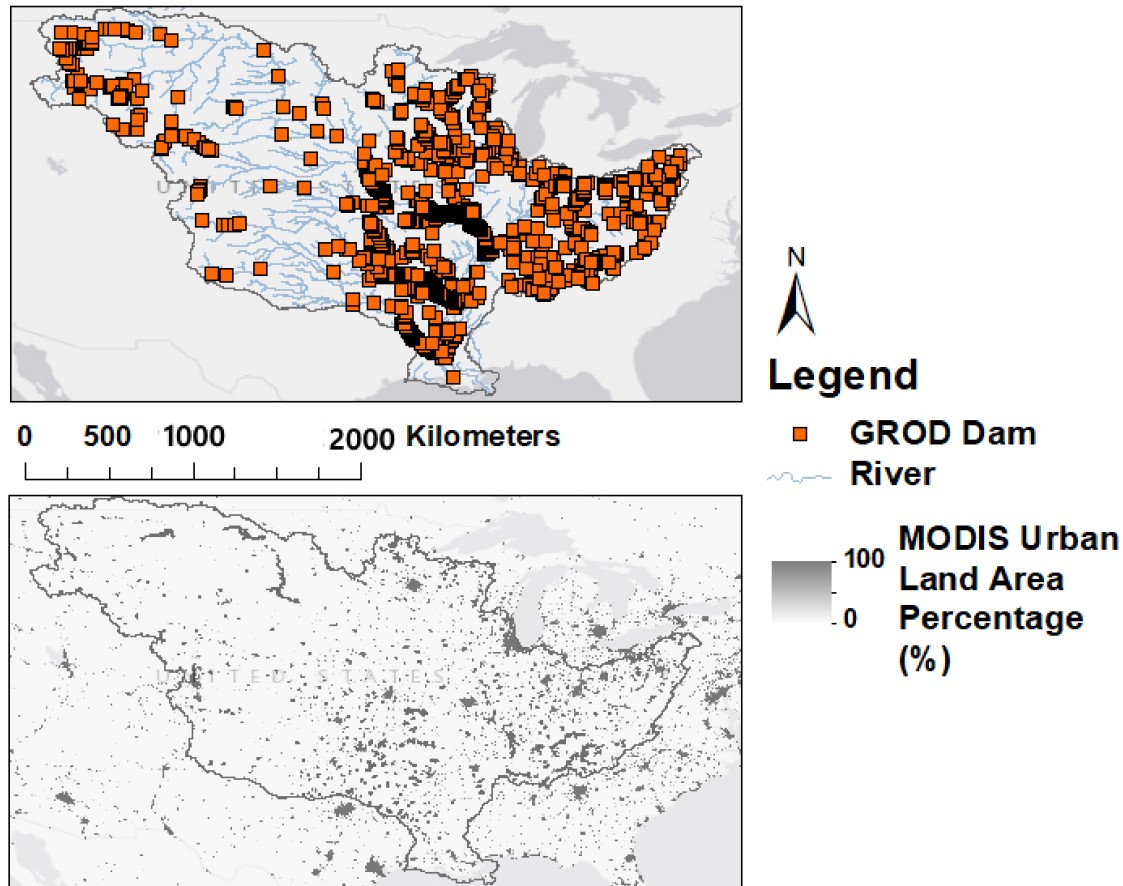

**Figure 3.** The Mississippi River Basin outline showing the Global River Obstruction Database (GROD)'s "Dams" class with MERIT Hydro rivers (top) and the MODIS "Urban and Built-up Lands" layer that gives a percentage of urban area per 0.05 degree of land area (bottom).

All code and text files used for the implementation of this research are available through CUAHSI's HydroShare (https://www.hydroshare.org/, accessed on 17 April 2021) with the following doi:10.4211/hs.7fcf864f87f546f090063f7dc1690920 [52].

### 2.4. Application Using Simulated SWOT Data

Our previous study [3] derives Log Pearson Type III discharge frequency distributions for 442 gauge locations in the Mississippi River basin for daily gauge time series and derived synthetic SWOT time series. The Log Pearson Type III distribution is highly recommended for use in the contexts of flood frequency analysis [53]. Here, the same is done with the subset of 373 gauges, determining which could be expanded within the limitations. Quantile (5%, 25%, 50%, 75%, and 95%) values are determined from each distribution and compared among each other. In the previous study [3], the relative root-mean-square-error (RRMSE) is used to compare quantiles, but for this paper, the KGE metric is selected to compare gauges. In addition, the Kolmogorov-Smirnov (KS) and Student $t$ tests are also utilized at the 95% significance level. Passing the KS test indicates the individual time series compared are drawn from the same continuous distribution and passing the $t$ test implies the means of the distributions are considered the same. Event peak flows greater than the 75th percentile per site and the number of days between the observed peak event flow and the nearest SWOT observation at that site are also identified.

For the application of the drainage area ratio method to SWOT observable rivers, synthetic SWOT time series accounting for the spatial and temporal irregularities of future

SWOT measurements are simulated. The method is introduced and used in [3] and [54], but summarized here. Three different time series are compared: daily discharge from USGS gauges (Qg), this gauge discharge sampled on days SWOT would observe the gauge (Qgs), and SWOT sampled discharge with added uncertainty (Qs). To simulate the synthetic SWOT time series Qgs, the 3-year span of daily gauge data is subset to only contain measurements for days SWOT would pass over each gauge site in the 21-day orbit cycle, assuming an initial orbit cycle starts on April 16, 2013. The data subset is determined by spatially joining a SWOT orbit shapefile [55] and the desired USGS gauge points to find what days SWOT would gather observations at each site. Then, to simulate Qs, uncertainty is added to these measurements considering the potential median −17% relative bias (rBIAS), and median 42% RRMSE associated with [45]'s SWOT discharge algorithm performance from 19 study rivers. The uncertainty for Qs is dependent on the selected discharge algorithm. These performance metrics are transformed into the log-space and used to estimate a log mean and standard deviation for a random Gaussian error distribution. RRMSE is representative of the spread or standard deviation of the data, while rBIAS is an indication of the under or overestimation of the data, similar to the mean. Each SWOT sampled discharge value obtained from the daily time series is then transformed into the log-space, the uncertainty from the lognormal Gaussian distribution added, and the discharge value is transformed back from the log-space. Representative magnitudes of error are better captured among low and high discharge values when the log-space is utilized.

For this study, in addition to comparing Qg with Qgs and Qs, it is also compared with expanded versions of both time series. Qgs and Qg are expanded using the drainage area ratio method with additional SWOT observations in neighboring reaches within the 0.01 to 100 ratio that do not have dams or a difference in more than 196 km$^2$ cumulative urban area between gauge locations. These time series are denoted as Qgs,e and Qs,e. To quantify the uncertainty of adding measurement points to each time series, a second expanded version of each Qgs and Qg exist, denoted as Qgs,e* and Qs,e*. These time series are expanded using their own discharge values rather than those donated with the drainage area ratio method on the same additional SWOT observation days. For gauge time series able to be expanded, KGE is calculated between distribution quantiles derived from the daily discharges and those derived from all simulated SWOT time series, original and expanded. The KS and Student *t* tests for these comparisons are also performed and previous peak flow analysis is replicated [3]. The number of days between recorded gauge peaks greater than the 75th percentile per site is found with and without expansion.

Since SWOT data products will give a single discharge value per reach, it is assumed that the amount of times SWOT will observe the river reach is equal to the number of times SWOT will observe a gauge on the same river reach. For instance, a SWOT pass may observe a section of a reach that the gauge does not lie on, yet because there is only one value associated with this reach, and the gauge does lie on the reach, it will be assumed that a discharge value for the gauge location will be obtained for that SWOT observation. Of the subset of gauges in the Mississippi River basin, this occurs for 26% of gauges. Additionally, it should be noted that the number of SWOT passes over a specific location could be larger than that recorded if a location is passed over twice on the same day. A daily scale for the time series is selected to match the time scale of the daily USGS gauge data.

## 3. Results

### 3.1. Criteria for Using the Drainage Area Ratio Method

To discern a possible limit in the range of acceptable area ratios, daily time series of the 354 participating gauges are compared. The amount of possible site donations from a donor gauge to a receiving gauge is 4344 within the set area ratio limits of 0.01 to 100 without accounting for whether the receiving gauge needs the specific SWOT day the donor gauge will provide (i.e., receiving gauge observed on same day as donor gauge). KGE is calculated between the donor gauge's time series multiplied by the scalar drainage area ratio and the

receptor gauge's true discharge time series. When plotting drainage area ratio vs. KGE for all possible donations (Figure 4a), there is an obvious increase in maximum performance as the ratio converges to 1, but no definite line of poor performance can be confidently stated. When the data are split into boxplot deciles (Figure 4e), with each bin representing 10% of the KGE results, all deciles have measurements that perform well (KGE > 0.3) and poorly (KGE < −0.41). In fact, for all donations upstream of the donor gauge (drainage area ratio <1), more than 75% of the data are above −0.41, yet all bins have values above the 75th percentile that are less than −0.41. In addition, the difference of drainage area between catchments is not a sure indication of performance either. Gauges that have less than 1000 km² of drainage area between them do not automatically give higher values of KGE than gauges with 100,000 km², as indicated by the color scales (Figure 4). There are less comparisons with a large difference in drainage area above the good KGE threshold (yellow points), but KGE values of all thresholds are still plausible no matter the difference in drainage area between compared gauges.

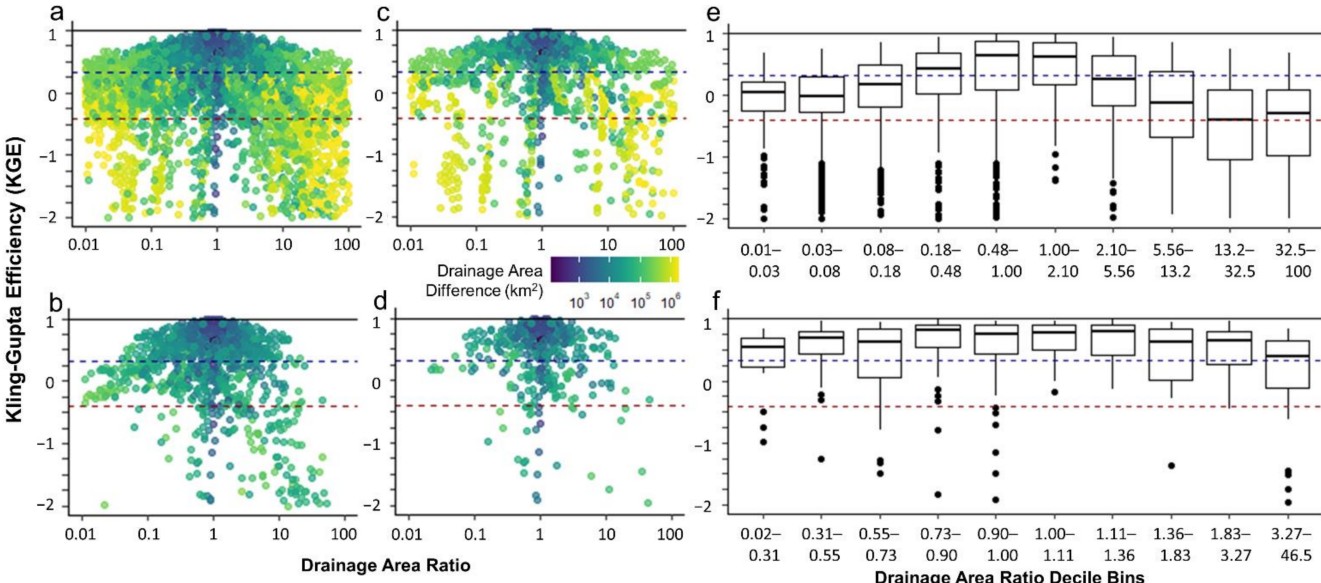

**Figure 4.** (**a–d**) Kling Gupta Efficiency (KGE) values between the 354 gauges vs. drainage area ratio. Points are colored by the difference in drainage area (km²) between the compared gauges. (**a**) all possible gauge comparison results (4344 pts), (**b**) gauge comparison results with cumulative urban land less than 196 km² between gauges (1340 pts), (**c**) gauge comparison results without GROD dams in between gauges (1412 pts) and (**d**) Filtered by comparison results without GROD dams or cumulative urban land area greater than 196 km² between gauges (510 pts). (**e,f**) KGE values compared via boxplot with decile bins of drainage area ratio. (**e**) Each boxplot has a tenth (~434) of the total results (4344). (**f**) Each boxplot has a tenth (~51) of the total filtered results (510). For all panels, the black horizontal line at 1 indicates the optimal KGE value, the blue dashed line indicates the study's defined "good" KGE value of 0.32, and the red dashed line is at KGE = −0.41, the threshold for the metric where the model improves upon the mean flow benchmark. Results are cut off at KGE = −2, but 563 comparisons (13%) are below this threshold for panels a and e, 82 (6%) for panel b, 113 (8%) for panel c, and only 16 comparisons (3%) are below the threshold for panels d and f.

Gauges located along rivers with a high degree of urbanization (i.e., large amounts of impervious area) or dams/reservoirs between donor and receptor gauges can account for a portion of the poor KGE values. If gauge comparison results likely impacted by urbanization are removed, quantified by an increase of at least 196 km² of Urban Built-Up lands between gauges (Figure 4b), 1340 of the original 4344 comparison location points (i.e., 31%) remain. This filtering removes all locations with a large difference in drainage area (>300 K km²). The maximum drainage area ratio resulting is 88, and the minimum is 0.011, but most points lie between 0.1 and 10. Removing comparisons likely impacted only by dams results in 1412 interactions, or 41% of the original possibilities (Figure 4c).

Unlike removing points because of urbanization, removing interactions because of dams identified by the GROD dataset still leaves comparisons within the entire range of difference in drainage area between gauges as well as the full range of drainage area ratios. It is intuitive that urbanization would filter out more comparisons because along large rivers, it is difficult to find river reaches in the United States whose cumulative drainage area is unaffected by cityscapes. Removing comparison results with greater than 196 km$^2$ of Urban Lands or dams between the sites (Figure 4d) results in 510 of the 4344 points, or 12% of the possible interactions. Splitting these points into boxplots of deciles (Figure 4f), the largest drainage area ratio when filtered is 46.5 and the smallest is 0.021, still a large range of ratios. Removing the first and last boxplots, 80% of the data lies between a drainage area ratio range of 0.31 and 3.27. All decile bins have at least 50% of their measurements above KGE 0.32, an improvement from Figure 4e with deciles of the unfiltered data comparisons. The portions of data considered "good" (KGE > 0.32) for each panel of Figure 4 are: 33% for all possible interactions (a,e), 58% when filtered by urbanization (b), 49% when filtered by dams (c), and 73% when filtered by both urbanization and dams (d,f). The portions of data that perform poorly (KGE < −0.41) for each panel of Figure 4 are: 32% for all possible interactions (a,e), 17% when filtered by urbanization (b), 25% when filtered by dams (c), and 9% when filtered by both urbanization and dams (d,f). Filtering by urbanization improves upon KGE metrics more than filtering by dams alone but filtering by both simultaneously greatly improves performance.

Figure 5 presents an example of the effect urbanization has on discharge (i.e., via changes in KGE values) for a stretch of river near Wichita, KS. The percent urbanization in the landscape from the MODIS satellite is displayed around six gauges with drainage area ratios between 0.86 and 1.16 in comparison to one another with actual drainage areas between 100,500 and 116,300 km$^2$ (Figure 5a). Figure 5b gives an example of the discharges at each gauge for the summer of 2014. The three upstream gauges above the city of Wichita give much lower discharges than the three downstream gauges, especially for the most upstream gauge that is also unaffected by runoff from the smaller adjoining city, Hutchinson, KS, contributing 24 km$^2$ of Urban Lands between USGS site numbers 07,142,680 and 07143330. When the drainage area ratio method is utilized, KGE values range from –9.6 to 0.7 (Figure 5c). Values are color coded by the difference in cumulative urbanization between gauges. For upstream gauge donations, meaning the donor gauge discharge downstream is multiplied by the drainage area ratio and compared to discharge at a location upstream, it is more obvious sites with the largest urbanization between them give poor performing KGE values, indicating less correlation, larger bias, and larger variability between discharges. In fact, a distinct difference in the spread of data exists for upstream vs. downstream donations (Figure 5d). All downstream donations result in comparable discharge time series, giving KGE values higher than the mean flow benchmark, −0.41. The same cannot be said for donations in the opposite direction. The discrepancy could be due to the way the KGE metric handles bias for comparisons with large differences in mean flow, as discussed in the discussion. The points in yellow and orange on Figure 5c are below the set threshold of 196 km$^2$ urbanization area defined in the Methods Section 2.2, yet some comparisons still result in KGE values less than −0.41, indicating that discharges are still significantly affected by the urban area below the defined threshold. Additionally, some gauges with a high difference in cumulative urbanization still give good KGE values, indicating high correlation, low bias, and low relative variability between gauge discharges, no matter the upstream or downstream direction of donation. These discrepancies within the defined limits are discussed in the discussion.

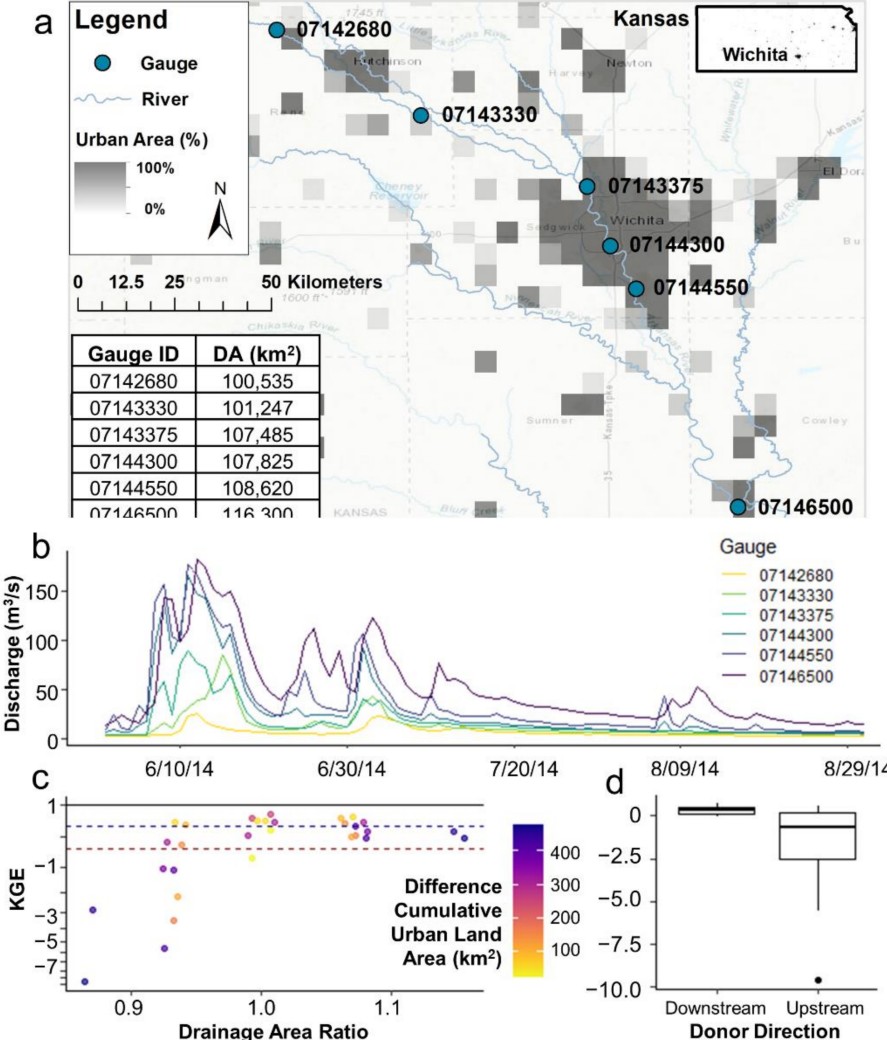

**Figure 5.** Case study of the effect of urbanization on discharge and Kling Gupta Efficiency (KGE) values. (**a**) Six gauges near Wichita, Kansas showing the percent of urban land area per 5 km$^2$ from MODIS satellite imagery in the area. USGS Gauge ID's and drainage area are recorded, and drainage area ratios between the gauges span from 0.86 to 1.16. (**b**) Example of daily discharge time series from June to August 2014 for each gauge record. (**c**) KGE vs. drainage area ratio for all gauge interactions colored by the difference in cumulative urban land area (km$^2$) between the compared gauges. The black horizontal line at 1 indicates the optimal KGE value, the blue dashed line indicates the study's defined "good" KGE value of 0.32, and the red dashed line is at KGE = −0.41, the threshold for the metric where the model improves upon the mean flow benchmark. KGE is mapped using a pseudo log10 scale. (**d**) Boxplots of the KGE metrics according to the direction of donation from the donor gauge.

Figure 6 presents an example of the effect dams and reservoirs have on discharge for six gauges on a stretch of river on the border of Colorado and Kansas. Two GROD dams are located along the stretch of river, shown by orange squares, one of which corresponds to a reservoir polygon also given in the GRanD database (Figure 6a). The drainage area ratios lie between 0.57 and 1.75 with actual drainage areas between 37,000 and 65,000 km$^2$. Figure 6b gives an example of the discharges at each gauge for May through September 2015. The effect the dams and reservoirs have on the discharges are obviously seen. The gauges with the lowest drainage area, upstream of the reservoir, have the highest amount of discharge for the first half of the summer months during the rainy season. There is a shift in July when the reservoir water is released through the dams and discharge downstream increases. The stark control of flow influences KGE values for comparisons of gauges

upstream and downstream of the dams (Figure 6c). KGE values are considered good if no GROD dams are between gauges, and when one dam is between the gauges, KGE decreases, and finally, when two dams are in between gauges, poorer KGE performance ensues. Like in Figure 5, there is a separation in the adequacy of performance for donations upstream vs. downstream, though in the opposite manner (Figure 6d). Donating downstream from a gauge above the dams to a gauge below the dams always gives a poor KGE, indicating poor discharge correlation, high bias and relative variability. This is intuitive because the drainage area ratio would multiply the discharge values to make them larger to compare to the downstream gauges, but the downstream discharges are lower than the upstream discharges in this case because of the dam influence. Alternatively, donating upstream with a scalar drainage area ratio multiplier that decreases the discharge further actually gives reasonable KGE values between −0.41 and 0.32 even with a dam in between. The discrepancy could be due to the way the KGE metric handles bias, as discussed in the discussion.

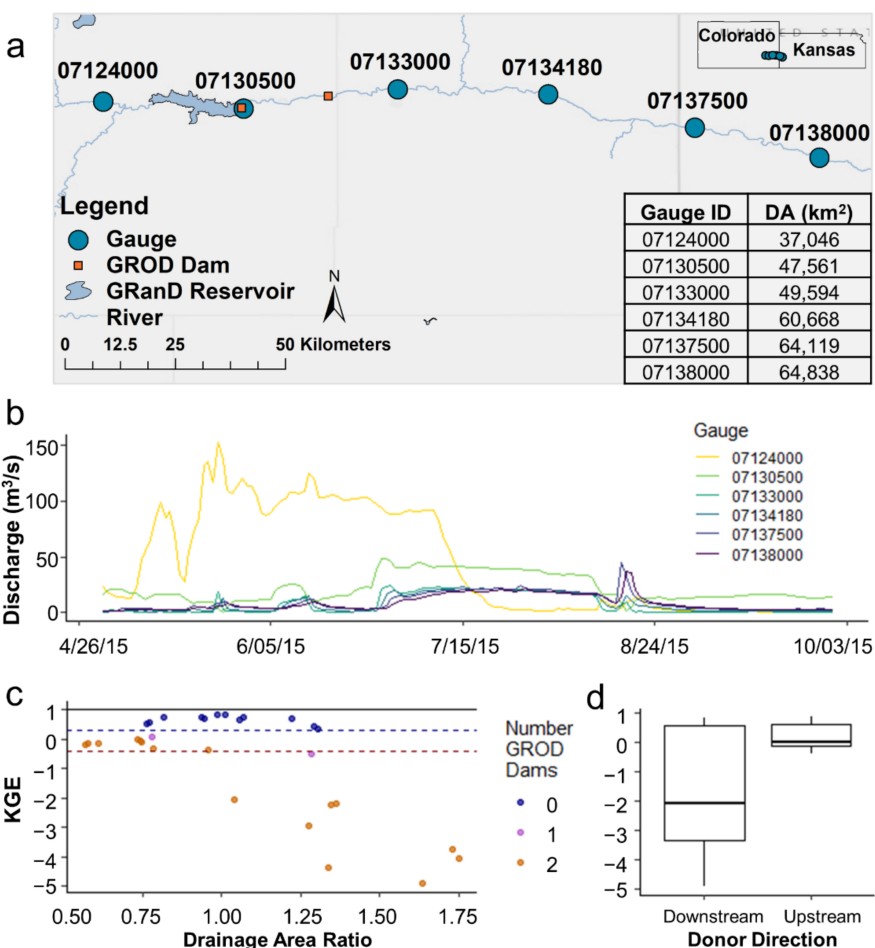

**Figure 6.** Case study of the effect of reservoirs and dams on discharge and Kling Gupta Efficiency (KGE) values. (**a**) Six gauges near the John Martin Reservoir spanning parts of Colorado and Kansas. Gauge ID's and drainage area are recorded, and drainage area ratios between the gauges span from 0.57 to 1.75. (**b**) Example of daily discharge time series from May to September 2015 for each gauge record. (**c**) KGE vs. drainage area ratio for all gauge interactions colored by the number of GROD Dams between compared gauges. The black horizontal line at 1 indicates the optimal KGE value, the blue dashed line indicates the study's defined "good" KGE value of 0.32, and the red dashed line is at KGE = −0.41, the threshold for the metric where the model improves upon the mean flow benchmark (**d**) Boxplots of the KGE metrics according to the direction of donation from the donor gauge.

To further justify the selection of criteria and to isolate the effect of each individually on results, the KGE information is displayed in panels of boxplots according to the number of GROD dams between gauges and quartiles of urban area observed between gauges using all possible results and results filtered by the alternate effect (Figure 7). Panels a and c clearly display the decrease in overall KGE performance the more GROD dams there are between gauges for all the potential interactions (Figure 7a) and those without the influence of urbanization above the defined threshold of 196 km$^2$ (Figure 7c). From panel a to c, the percent of interactions without any dams at all increases from 32 to 38%, the boxplot with 1–2 dams increases from 30 to 32% of the data, the 3–5 dams category decreases in percentage from 22 to 14% of the data, and the greater than 5 dams remains the same at 16% of the data for both. By eliminating the datapoints above the urban land area threshold, gauge comparisons with 3 or more dams are represented less in the dataset. Solely observing the influence dams have on method performance, the median KGE decreases by 55% from 0.66 to 0.3 when any number of dams are between gauges. Panels b and d also highlight a decrease in KGE performance correlating to greater differences in cumulative urban area between gauges for all data (Figure 7b) and those without the influence of dams (Figure 7d). Each of these boxplot pairs are separated into quantiles, representing 25% of the data. For the determined 196 km$^2$ threshold, without the influence of comparisons affected by dams, the median KGE for comparisons below the threshold is 0.67, while above the threshold, the median KGE is 0.04, a decrease in performance by 94%. For panels with all results represented (Figure 7a,b), there is a distinct increase in performance for upstream gauges rather than downstream gauges. On the other hand, in Figure 7c, the first two boxplot categories have quartiles that behave similarly regardless whether they are separated by upstream or downstream donations. Ridding the data of large amounts urban land area between gauges results in more uniformity of performance for donor direction. The same can be said for Figure 7d; without the influence of dams, there is no obvious difference in performance by donor direction until the last set of boxplots with urbanization higher than the 75th percentile. Notably, large differences in the amount of urban area between gauges show a distinct difference in upstream and downstream donations.

### 3.2. SWOT Time Series Expansion Application

Now, the potential increase in SWOT observations throughout the basin are analyzed, not just potential donations between gauges. All reaches within the Mississippi River basin are classified by the number of observations within SWOT's 21-day orbit cycle, counting only one pass per day if there were multiple passes in one day (Figure 8a). The median number of observations is two with a maximum of four per orbit (Figure 8d). Drainage area ratios within the 0.01 to 100 range are used and potential gauge donations filtered out that have greater than 196 km$^2$ of urban area between gauges and/or any GROD dams between gauges (Figure 8b). With expansion, the median number of observations per reach increases by 50% to three, with a maximum of sixteen observations per orbit cycle (Figure 8d). The spatial change in the number of observations when the SWOT time series are expanded using the drainage area ratio method with the limitations vs. the original number of observations is shown in Figure 8c. The number of observations for a given reach has a mean increase of 83%, providing a median of three estimates per 21-days. In total, 63% of reaches increase their number of observations with expansion. The western portion of the Mississippi River basin increases in the number of observations far more than the east. The west is not as impacted by urban infrastructure as the east, and a higher density of urban areas exists in the east. The set limitations for the drainage area ratio method's use therefore limits the eastern portion of the Mississippi river basin far more than the west. With the opportunity of expansion, 37% of reaches have no change in the number of observations per orbit, 28% of reaches increase by one observation, 18% increase by two, and 17% increase by three or more observations. Not only are the median number of observations two and three for maps (a) and (b), respectively, but they are also the

mode number of observations. Originally, 46% of reaches would have been observed twice, decreasing to 20% with the expansion (i.e., because an additional 26% of reaches are now observed 3+ times).

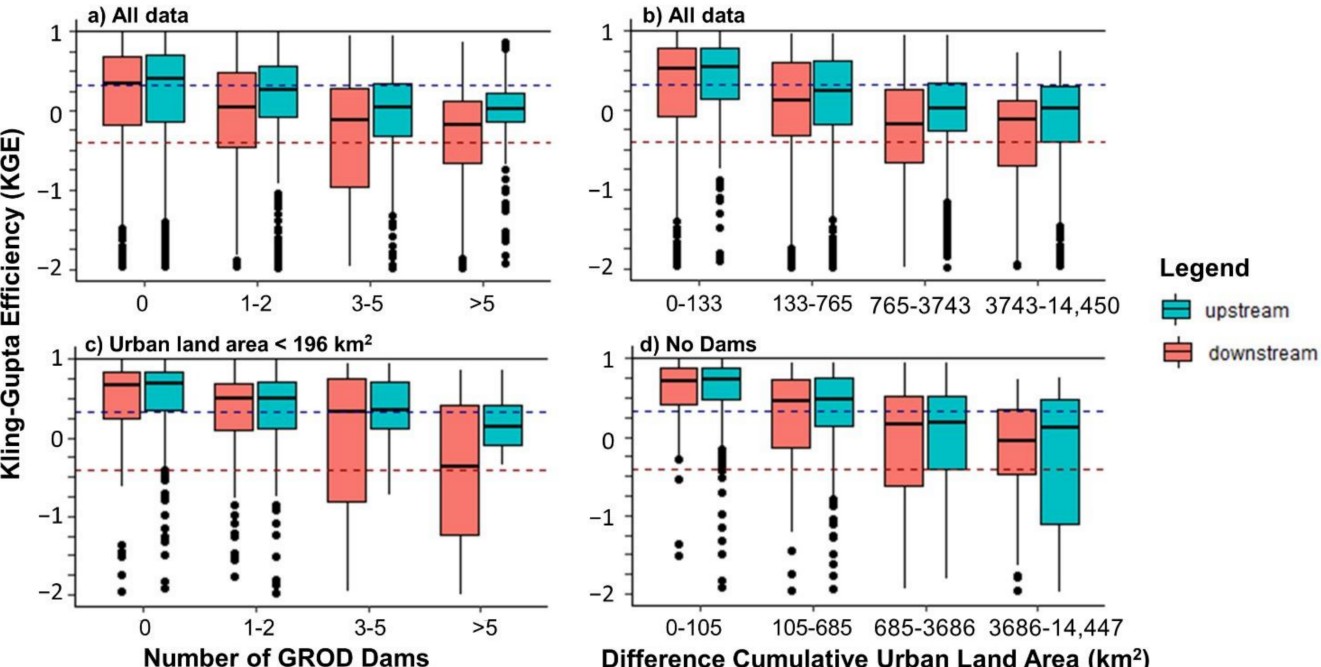

**Figure 7.** Boxplots for (**a**) the number of GROD dams per gauge comparison for all 4344 potential donations, (**b**) the difference in cumulative urban land area between gauges for all 4344 potential donations with each boxplot representing 25% of possible results, (**c**) the number of GROD Dams per gauge comparison for only the 1340 results with less than 196 km² of urban land area between gauges, and (**d**) the difference in cumulative urban area between gauges for only the 1412 results without any GROD dams between gauges with each boxplot representing 25% of possible comparisons. For all panels, the black horizontal line at 1 indicates the optimal KGE value, the blue dashed line indicates the study's defined "good" KGE value of 0.32, and the red dashed line is at KGE = −0.41, the threshold for the metric where the model improves upon the mean flow benchmark. Boxplots are split into donations upstream vs. donations downstream between gauges.

Returning to the gauge analysis, of the 373 USGS gauges, discharge series for 167 gauges can be expanded within the limitations, receiving a different SWOT measurement on a day not observed at its location. There are 390 donations in total among the 167 gauges. One donation is equivalent to gaining one day of observation within the 21-day orbit cycle, therefore multiple donations can exist from one gauge to another. For all time series distributions, only these 167 sites were used to calculate KGE values for the quantiles and to conduct the KS and Student *t* tests (Table 1). For ease of discussion, the six comparisons are categorized into two groups: the Qgs group, Qg comparing with a form of Qgs in the top half of the table, and the Qs group, Qg comparing with a form of Qs shown in the latter half of the table. The same pattern of results persists for all quantile comparisons in each group. The Qgs group, accounting for only SWOT sampling, gives better results than the Qs group, when uncertainty is added to the sampling. Quantile comparisons with time series expanded using the drainage area ratio method (Qgs,e and Qs,e) generally perform slightly worse than their counterparts, except for the higher flows, the 75th and 95th percentiles. Adding more values increases the chance that higher flow values will be captured, better reflecting true high flow quantile values. Regardless, in each group, KGE values are similar to each other whether the comparison is between the original synthetic SWOT time series, the drainage area ratio expanded time series, or the SWOT time series expanded using its own location's discharge on the added days. Among these three comparisons, the greatest difference in KGE performance for a quantile is only 0.09. For the significance tests, passing the KS test indicates the samples of data are derived from

the same continuous distribution and passing the Student *t* test indicates the means of the samples are statistically the same. For the KS test, surprisingly, the highest percentage of sites passing the test had no expansion at all. Even expanding the time series by discharge values of its own series to represent the additional days performs slightly worse. This could be by chance, with the additional SWOT days skewing the quantile data high or low depending on the sample for some gauges, though relatively unaffecting the median (Q50). For the *t* test, the highest percentage from the Qgs group was Qg vs. Qgs,e*, as expected, but for the Qs group, Qg vs. Qs gave the highest number of sites passing. Increasing the amount of measurements may not perform as well when the time series has uncertainty, giving more opportunity for means to differ. For both groups, the expanded distribution comparisons using the drainage area ratio method performed the worst for both tests. Adding values scaled by drainage area ratio increases the uncertainty, decreasing the number of sites where compared distributions and means are statistically the same.

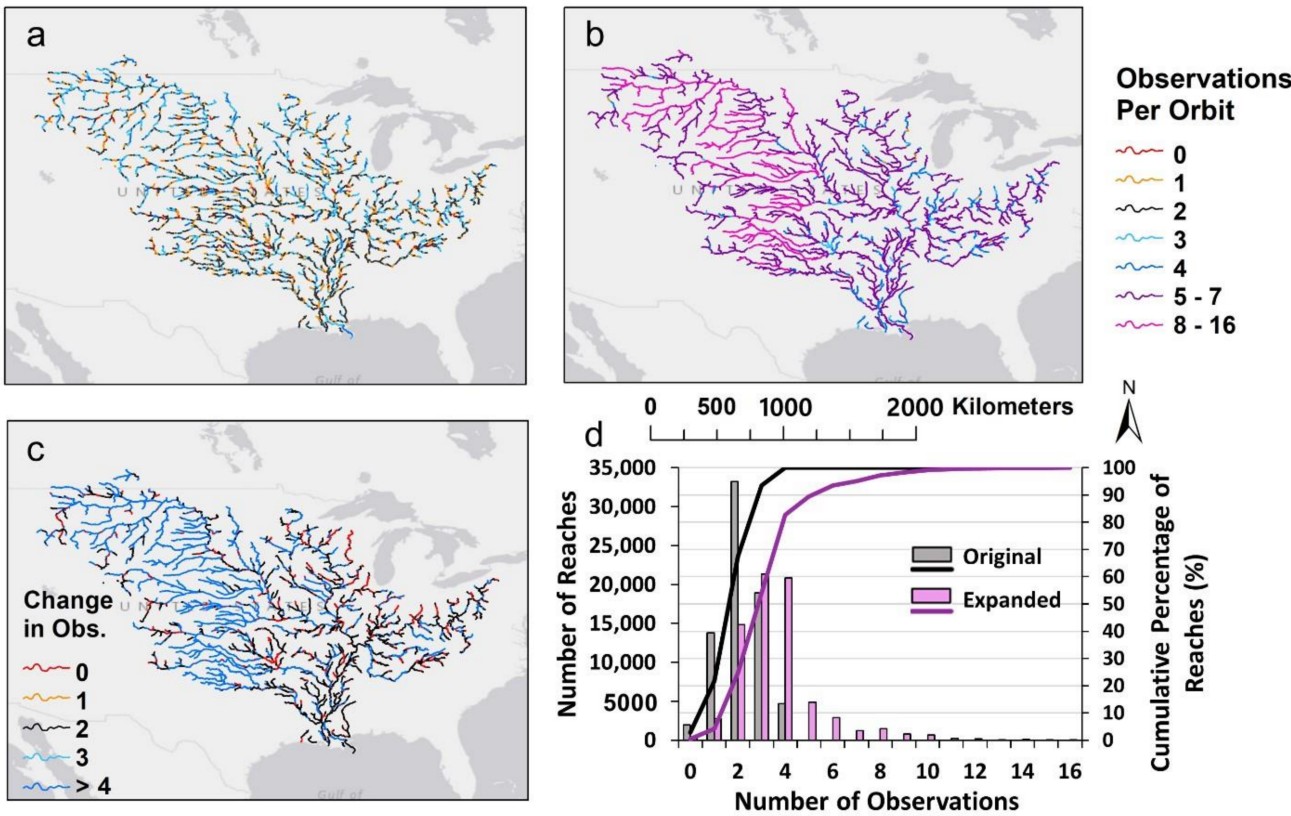

**Figure 8.** All study reaches in the Mississippi River basin classified by (**a**) the original number of SWOT observations per 21-day orbit cycle without drainage area ratio method expansion, (**b**) the number of SWOT observations per 21-day orbit cycle with drainage area ratio method expansion for ratios between 0.01 and 100, using only donations without dams and urbanization less than 196 km$^2$ between reaches, and (**c**) the change in observations between (**a**) and (**b**). (**d**) The number (bars) and cumulative percent (lines) of reaches observed categorized by the number of SWOT observations per orbit. Color indicates which map the statistics describe. Black gives the original quantity of reaches that are observed a specific number of times by SWOT (**a**) and purple represents that of the expanded study quantities (**b**).

Replicating the peaks analysis from [3] reveals a drastic improvement in diminishing the number of days between when a peak is observed and when SWOT collects a measurement when discharges are expanded (Figure 9). There is a 100% increase in capturing a measurement on the exact day when a peak storm event occurs, increasing from 10% to 20% of peaks greater than the 75th percentile for each site. Originally, 56% of peaks have sampled SWOT measurements within three days of the measured gauge event, and this percentage increases to 81% with expansion. In addition, 99% of peak measurements have

a SWOT observation within five days of the peak event with expansion, while only 80% can be said for the original SWOT sampled measurements.

**Table 1.** KGE values for distribution quantiles compared between the daily distribution (Qg), and those of the SWOT simulated time series for the 167 gauges able to be expanded by additional SWOT measurements. Qgs and Qs distributions are derived from the original SWOT sampled time series, with Qs also including uncertainty. Qgs,e and Qs,e indicate expansion using the drainage area ratio, and Qgs,e* and Qs,e* indicate expansion with the gauge's own discharges on added SWOT observation days for comparison purposes. The percentage of 167 that passed both the Kolmogorov-Smirnov (KS) test and Student *t* test for each distribution comparison are also listed.

| Comparison | KGE Q5 | KGE Q25 | KGE Q50 | KGE Q75 | KGE Q95 | % Pass KS Test | % Pass *t* Test |
|---|---|---|---|---|---|---|---|
| Qg vs. Qgs | 0.94 | 0.97 | 0.98 | 0.98 | 0.88 | 100 | 90 |
| Qg vs. Qgs,e | 0.92 | 0.96 | 0.98 | 0.99 | 0.96 | 60 | 65 |
| Qg vs. Qgs,e* | 0.97 | 0.98 | 0.99 | 0.98 | 0.94 | 98 | 99 |
| Qg vs. Qs | 0.63 | 0.72 | 0.78 | 0.81 | 0.81 | 91 | 67 |
| Qg vs. Qs,e | 0.54 | 0.69 | 0.78 | 0.84 | 0.88 | 51 | 46 |
| Qg vs. Qs,e* | 0.61 | 0.70 | 0.77 | 0.82 | 0.87 | 83 | 66 |

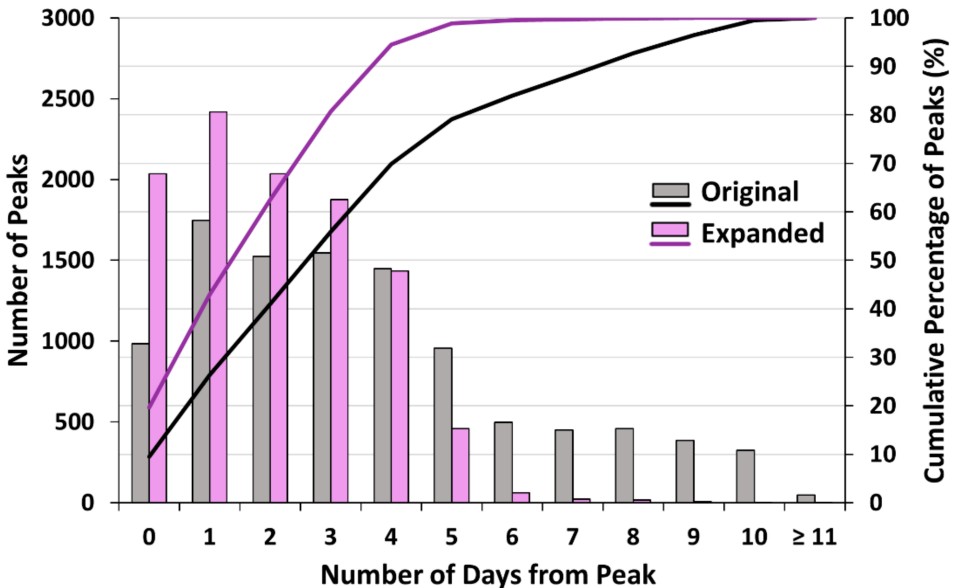

**Figure 9.** The number of peak flows greater than the 75th percentile for daily gauge time series and the corresponding number of days between the event peak and a SWOT observation for both the original and expanded time series. The count and cumulative percentage of each are displayed.

## 4. Discussion

### 4.1. Drainage Area Ratio Method Criteria

The discharge donations between reaches on SWOT observable gauges cannot be conclusively limited within the range of drainage area ratio limits tested (0.01 to 100). The closer the ratio to one, the more opportunity for good results, but "good" results were observed across the range of ratios (Figure 4a,e). Similarly, success in utilizing the drainage area ratio method has been found for ratios as low as 0.005 [39]. Like other studies [31,34], extreme variability in the method's performance is found in this study over all ranges of tested ratios. In an ideal river basin without anthropogenic influence, it is plausible that for SWOT observable rivers, if one river location flows into the other, regardless of the ratio of drainage area, there could be a reasonable donation of flow from one to the other using the drainage area ratio method. Less emphasis should be placed on limiting the range of drainage area ratios and more on how the assumption that discharge scales linearly with drainage area is potentially violated in a region. Unfortunately for regionalization techniques, it can no longer be assumed in the ever-urbanizing environment that large

rivers akin to the magnitude SWOT will observe (>50–100 m wide) are unaffected by anthropogenic effects. Discharge does not scale linearly with drainage area in regions where man-made infrastructure is pervasive. However, when limits are placed on utilizing the drainage area ratio method for regions influenced by dams and urban land area, 80% of filtered results have drainage area ratios between 0.31 and 3.27, similar in magnitude to the drainage area ratio limits mentioned in the introduction deemed as acceptable by other studies [32,37,38].

To utilize the drainage area ratio method most effectively with SWOT observable rivers, no dams should be between locations, and the potential influence urban areas have on runoff should be considered. In this study, to consider the impact of urban areas, an estimated limit of 196 km$^2$ is placed on the amount of acceptable urban land area between potential locations. Cities' effect on metrics may not be accurately quantified solely by this limit imposed on MODIS' "Urban Built-Up Land" layer between gauges, however. The point in Figure 5c that has 24 km$^2$ of urban land area between gauges, yet still gives a KGE value less than −0.41, is an upstream comparison between gauges 07,142,680 and 07143330. This datapoint is not filtered out with the criterion and thus highlights that the threshold of the amount of urban land area between locations is imperfect. Smaller cities directly in between gauges could have a large impact on KGE. Other areas with the same amount or greater cumulative urban area between gauges still produce reasonable KGE values though. The key factor could be the location of urban land in the additional catchment area added between gauges. If the city is directly in-between locations along the river, there is a higher chance that the discharge could be significantly influenced by urban area. If a threshold of no urban area between gauges is imposed though, 99% of comparisons would be considered void. A different attempt was made to characterize urbanization as the difference in cumulative percent urban lands between gauges in this study, but a conclusive threshold could not be found with this classification. Another study uses similar criterion when selecting appropriate donor gauges in the Susquehanna River Basin but quantifies the urban impact differently [56]. They state that streamflow at a gauge should not be significantly affected by upstream regulation, diversions, or mining, and a donor gauge should have less than 15% urban area in its watershed. SWOT observable rivers in the United States will most likely be affected by urbanization, therefore, sharing data among neighboring reaches on a SWOT observable river with a scaling factor may not adequately capture the discharge dynamics.

Filtering by the presence of dams or an increase in urbanization in-between gauges also unfortunately removes many locations where reasonable KGE values are obtained. Not all dams are always highly controlled, thus using the drainage area ratio method from one side of a dam to another could give reasonable results depending on the level of control. Additionally, in Figure 5c, though gauges 07,144,550 and 07,144,300 have a difference in cumulative urban area of 253 km$^2$, higher than the threshold, their KGE values when compared to one another are good: 0.7 for the downstream and 0.59 for the upstream donation. Though a significant amount of urban land area is added between the two gauges, both points are in the middle of the city and therefore likely being affected by the same increase in discharge. Discharge at these locations behaves similarly, still producing correlated discharge and high KGE values. It is to be noted that this may not be the case though for all gauges that are in the same city; differing runoff outfall locations and how/where water drains into the rivers could be another underlying factor out of the scope of this study. Recently, a method has been developed for accounting for the influence of anthropogenically modified landcover that incorporates stormwater sewer networks and retention basins to reduce water budget errors in urban systems [40]. Incorporating anthropogenic influences in this or other ways for further regionalization methods is advised.

### 4.2. Kling-Gupta Efficiency (KGE) Limitations

The large discrepancy in this study's KGE performance for donations upstream vs. downstream can be attributed to the bias ratio term in the KGE metric. When the bias ratio term is calculated, if there is a large discrepancy in the means of the data, since the bias ratio term is simply a ratio of these means, values can differ greatly depending on donation direction. For example, if one gauge produces a magnitude of mean discharge six times larger than the other, the bias ratio term could be 6 or 0.16 depending on which gauge is the donor gauge. In the case of dams and reservoirs impacting the volume of flow, a gauge donating downstream would end up with a much larger bias ratio, and therefore a much worse KGE value than the same gauges donating in the upstream direction. It is recommended that instead of taking a holistic approach to KGE values, its individual components be analyzed [44]. Modifying the metric through weighting of components may be a better approach to align with a study's purpose. When weights are applied to the bias ratio term, KGE more accurately calibrates a hydrologic model [57]. A weighted KGE could be further explored in future studies. One could question whether using the Nash-Sutcliffe Efficiency (NSE) would be better for this application, but NSE often underestimates observed flow variability, a term that KGE strongly considers. The perfect error metric does not currently exist.

### 4.3. Other Possible Issues with the Main Assumption

Additionally, note that the assumption that discharge scales linearly with drainage area can be challenged by a myriad of factors, not all of which are anthropogenic. A study is performed in a small boreal catchment with little anthropogenic influence in Northern Sweden [34], and still, the drainage area ratio method is not satisfactory in estimating flow. They cite seasonality, vegetation, and geology influenced landscape-stream connectivity and runoff magnitude, as well as topography differences as potential reasons for the poor results. Land cover type affects the amount of infiltration and therefore the speed at which water enters the main channel. In addition, not all parts of a river basin that flow into the river experience the same volume of precipitation and evapotranspiration, so water is not delivered uniformly to the river. It is also well cited that nearby catchments with seemingly similar landscape forms do not always behave similarly [58,59]. The effect of these other factors could be further analyzed in the context of larger rivers akin to those that will be observed by SWOT.

### 4.4. SWOT Time Series Expansion

In applying the drainage area ratio method within the defined criteria, previous findings [3] are successfully replicated: SWOT temporal sampling still has minimal impact on derived discharge frequency distributions. In the previous study, it is found from 442 gauges that 78% of distributions from Qs are statistically the same as that from Qg. For this study, the analysis from the previous study is redone in addition to the analysis with expansion because only 167 gauges that can expand their time series using the drainage area ratio method are analyzed, a subset of the original 442 gauges [3]. The percentage of Qs distributions considered to be statistically the same as Qg distributions increases to 91%, due to the decrease in sample size. When Qs time series' are expanded, that percentage falls to 51% and 83% for Qg vs. Qs,e and Qg vs. Qs,e* respectively. The same pattern occurs for the Student *t* test. Qg vs. Qs sets a baseline of 67% passing the Student *t* test, meaning their means are statistically considered the same, but this percentage falls to 46% and 66% for Qg vs. Qs,e and Qg vs. Qs,e* respectively. Expanding the time series does not improve upon previous results; the increase in data decreases the similarity between daily and sampled time series according to the KS and Student *t* tests. Adding neighboring values scaled by drainage area introduces more uncertainty, and thus gives poorer performance (Qg vs. Qgs,e and Qg vs. Qs,e). In the case of adding additional measurements from its own time series (Qg vs. Qgs,e* and Qg vs. Qs,e*), the significance test results do not decrease as drastically, but there is still a slight decrease, indicating that perhaps the expanded

discharges were not representative of the full time series in some cases. The data would not be more skewed, however, because according to the Student *t* test, the percentage of statistically the same means only dropped by 1%. For the quantile comparisons, though the expanded time series did not surpass the performance of the original simulated SWOT time series either, all KGE values are considered "good," with the lowest KGE value being 0.54 for the low flow quantile, the 5th percentile, congruent with previous results [3]. What was improved, however, was the potential to observe peak flow values by SWOT on the same day they occur. The percentage of same day observation increased from 10% to 20% with expansion, an increase of 100%. Only 167 of the 373 gauges can be expanded, not well representing the actual potential of data expansion. Only reaches that also have gauge discharges in the sample can be compared. When SWOT products are available for each 10 km observable reach, the number of measurements to draw from greatly increases.

In the context of expanding the temporal and spatial resolution of SWOT products, other more complex methods have recently been published. Data assimilation methods to generate a spatially and temporally continuous SWOT discharge data product across the Upper Mississippi River Basin network have been developed [29]. They derive discharge along 17 reaches (varying from 80 to 1095 km in length) according to the LISFLOOD-FP hydrodynamic model specifications, incorporating the temporally and spatially irregular SWOT simulated time series. They conclude that the more reaches are observed during the 21 days, the more accurate the modeled discharge, yet discharge accuracy improvement persists during the period between two SWOT observations. Similarly, data assimilation has also been used by combining the Mass-conserved Flow Law Inversion (McFLI) approach (as developed by [45]) for satellite discharge estimation and a globally forced hydrologic model in the Missouri River basin [60]. They find positive improvements on metrics for 92% of originally poorly modeled gauges on a daily scale. These more intricate approaches to expanding SWOT, or remote sensing time series in general, have great potential because they produce daily time series. The applicability of data assimilation via Kalman filtering with the potential for use by upcoming river-observing satellite missions [61] and inverse routing methods for spatiotemporal assimilation that could be applied to future SWOT products have also been explored [62,63]. To increase applicability, utilizing the method of incorporating anthropogenic impacts on a river network [40] could be integrated into models for greater accuracy. The ability to adjust discharges based on known infrastructure and urbanization influences is increasingly necessary.

## 5. Conclusions

To expand the spatial and temporal extent of SWOT derived discharges in the Mississippi River basin, limitations for the drainage area ratio method's optimal use are identified. No conclusive evidence is found to limit the method within a certain range of ratios, but it is recommended limitations be placed in regions where the assumption that discharge scales linearly with drainage area is likely not valid. Specifically, these regions are characterized as having river flow altered by surrounding urbanization or flow obstructed by instream infrastructure such as dams and reservoirs, which can negatively affect the performance of the drainage area ratio method. For large scale SWOT observable rivers (i.e., river catchment areas >1000 km$^2$) in the United States, it can be assumed urbanization and man-made infrastructure such as dams greatly impact the success or failure of the drainage area ratio method's performance. A 55% decrease in median KGE is found when any dams are present between two locations in the Mississippi River Basin, excluding comparisons affected by urbanization. Isolating the effect of urban land area, the median KGE value decreases by 94% for comparisons above the defined land area threshold vs. those below. For this study, two limitations for the drainage area ratio's use are proposed: (1) no dams or reservoirs between a donor location and the desired unknown location, and (2) the impact of urban landscapes on differences in flow upstream and downstream be considered. To quantify the latter, a limit of 196 km$^2$ of urban land area is allowable between locations in this study, but as discussed previously, other options should be explored.

Applying these criteria to this subset of gauges in the Mississippi River basin naturally limits drainage area ratios to between 0.02 and 46.5, with 80% of interactions between gauges with ratios between 0.31 and 3.27. Instead of discounting reaches as "unusable" for regionalization techniques if impacted by urbanization, it would be ideal to incorporate these increases of flow into the network topology of the basin. Regions with urbanization are often large population centers, and therefore, represent areas of vital interest to water resources managers. Not only do ideal discharge dynamics need to be understood, but also realistic dynamics in the ever-changing landscape.

Applying the criterion to SWOT observable rivers in the Mississippi River basin increases the mean number of observations for a given reach by 83%. In total, 63% of reaches increase their number of observations per 21-day orbit cycle with expansion. The number of peak events greater than the 75th percentile captured also increased by 100%, from 10% to 20% of events. Though the drainage area ratio method has the capability to increase the number of observations per reach, for the subset of 167 reaches with gauges studied, expanded SWOT sampled time series distributions often slightly underperform for quantile and significance test results in comparison to the performance of the original sampled time series, though KGE quantile results are still considered good. The method is applicable to other big scale river catchments and similar results would be expected. Future research could repeat these methods with a larger sample size, using modelled discharge available for all SWOT observable reaches in the Mississippi River Basin (70,000+) to fully explore the drainage area ratio's applicability instead of the small subset of gauges. Additionally, other time series expansion methods (e.g., regionalization, data-assimilation, inverse routing) to integrate derived SWOT discharges in space and time throughout a river network need to be explored.

Satellite discharge products time series expansion has the potential to better represent discharge dynamics, more accurately informing hydrologic applications. SWOT satellite discharge products are not meant to be a replacement for in-situ gauge networks or global modelled discharge estimates, but rather a compliment. Data assimilation methods often combine methods for more accurate estimations. For instance, informing hydrologic models with temporally or spatially irregular satellite data shows much promise [29,60]. Hydrologic models informed and calibrated using event discharges derived from SWOT can be used to predict specific flood occurrences and extents and to assess climate change impacts globally. Enabling better forecasting of extreme events like droughts and floods helps reduce potential losses if hazardous areas are appropriately identified in regions. River routing and water cycle processes are also incorporated into Global Climate Models (GCMs), which provide future projections of the changing world, estimating most probable ranges of impacts from natural and anthropogenic causes. Incorporating SWOT discharge products with other estimations from gauged in-situ data and hydrologic models can increase knowledge and help societies make more informed decisions regarding water management, climate change, and safety precautions.

**Author Contributions:** Individual contributions are as follows: Conceptualization, C.N. and E.B.; methodology, E.B. and C.N.; software, C.N. and E.B.; validation, C.N. and E.B.; formal analysis, C.N.; investigation, C.N.; resources, C.N. and E.B.; data curation, C.N.; writing—original draft preparation, C.N.; writing—review and editing, C.N. and E.B.; visualization, C.N.; supervision, E.B.; project administration, E.B. and C.N.; funding acquisition, C.N. All authors have read and agreed to the published version of the manuscript.

**Funding:** This research was funded by the National Science Foundation Graduate Research Fellowship Program, Grant No. 1451070. Any opinions, findings, and conclusions or recommendations expressed in this material are those of the authors and do not necessarily reflect the views of the National Science Foundation.

**Institutional Review Board Statement:** Not applicable.

**Informed Consent Statement:** Not applicable.

**Data Availability Statement:** All code and text files used for the implementation of this research are available through CUAHSI's HydroShare (https://www.hydroshare.org/, accessed on 17 April 2021) with the following doi:10.4211/hs.7fcf864f87f546f090063f7dc1690920 [52]. SWOT orbit shapefiles can be obtained here: https://www.aviso.altimetry.fr/en/missions/future-missions/swot/orbit.html, (accessed on 18 May 2018) [55], the GROD dataset can be accessed here: https://figshare.com/articles/dataset/GROD_US_and_validation_data/12003513 (accessed on 11 February 2021) [41], the GRanD dataset here: http://globaldamwatch.org/grand/ (accessed on 11 February 2021) [50], the MODIS dataset here: https://lpdaac.usgs.gov/products/mcd12c1v006/ (accessed on 19 February 2021) [49], the MERIT Hydro dataset here: http://hydro.iis.u-tokyo.ac.jp/~yamadai/MERIT_Hydro/index.html (accessed on 12 January 2021) [48], and USGS gauge discharges here: https://waterdata.usgs.gov/nwis/dv/?referred_module=sw (accessed on 10 January 2021).

**Conflicts of Interest:** The authors declare no conflict of interest.

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
