# Peer review of "Leveraging River Network Topology and Regionalization to Expand SWOT-Derived River Discharge Time Series in the Mississippi River Basin"

_remotesensing, doi:10.3390/rs13081590_

Round 1
Reviewer 1 Report
I find this study very interesting; it has the potential to provide insight into the contribution of expanding the spatial and temporal resolution of SWOT products.
While the field of research is interesting, I feel that description of your findings could be more comprehensive. Please find below a list of remarks regarding the manuscript content, which I feel should be addressed before publishing of your work and help to clarify some of the findings that you presented.
- Although many authors write in the first person, I still adhere to the classic impersonal writing, that is, in the third person. Please correct the whole article by replacing first-person pronouns with third-person pronouns.
- Keywords: Too many keywords, and please do not use the abbreviations.
- I suggest that the discussion section should be divided into several points and add the sub chapter titles, so that readers can understand the content of the discussion more clearly.
- Success to authors. Do not give up trying to publish this article.
Reviewer 2 Report
The paper is indeed interesting and I congratulate the authors for the high quality of the work done.
I suggest the following comments to the authors:
- In relation to the limit of 196km2 of urban land area used in the proposed method between 2 gauges, ¿should the urbanization area be treated as a percentage (%) of the total catchment area rather than a fixed constant number between 2 gauges?. It will not be the same the impact of having 196km2 of brownfield area in a 1,000km2 catchment area (aprox. 20%) than in a 2,000km2 catchment area (aprox. 10%).
- Title- Add the following: “Leveraging river network topology and regionalization to expand SWOT-derived river discharge time series in the Mississippi River Basin”
- LN11: Add “in natural conditions” or “greenfield conditions”: “It is often assumed discharge upstream and downstream of a river location are highly correlated in natural conditions”
- LN12: Add “such as dams and reservoirs” after instream infrastructure
- LN12: modify “impact this assumption” to something more specific (I would probably suggest to change it to “can invalidate this assumption due to the change in natural hydrological behaviour” or “can negatively impact” or similar)
- LN21: “significantly impacts its applicability” – need to be more specific
- LN23: “with limitations” – need to be more specific
- Abstract: Compare the main findings of the Conclusion section with the Abstract. Both need to be the same
- Abstract: try to include the main applications of this method and recommendations for future research/improvement of results
- LN95-LN106: The text includes the drainage ratios results found in different studies. For context and comparison, please add the catchment size range with which the different studies were working with and the urban area (since for this particular work, we are mainly interested in results for large scale river basins)
- Materials and methods – need to include a brief description of the Mississippi River Basin main characteristics (even if this is a map at big scale): geology of the river basin (infiltration areas), land use cover (agricultural/urban, etc), location of dams/reservoirs (main infrastructure), altitude, mean precipitation, etc.
- LN220: “gauges that have a record of at least 20 years” – is this a sufficiently long time series to support statistical variability? Add a supportive reference
- The selected period for analysis is from April 16, 2013 to April 15, 2016 – why is this period representative? Describe hydrologically this period of time (any droughts/floods, or normal years)
- LN236: Figure 2 (a) – change “1,000” for “1,100” (in accordance with LN234)
- LN254: Justify why the Log Pearson Type III was used (were there any other distributions used?)
- LN723: “large degree of urbanization” – please quantify “large”
- LN725: “large scale SWOT observable rivers” - please quantify “large” (river catchments areas larger than 1,000km2?)
- Compare the main findings of the Conclusion section with the Abstract. Both need to be the same
- Conclusions – quantify the influence of urban area and number of dams on the accuracy of the method
- What is the most suitable water field of application of this method (flood forecasting, climate change assessment, hydrologic modelling at big scale, etc.) and how could this method be combined/complimented with other methods or available gauged onsite data to improve/refine results?
- Could this method be applicable to other big scale river catchments and similar results be expected?
- Please state how these results could be integrated with future climate change projections to better forecast extreme events (droughts and floods)
- Future research directions need to be more specific
